# ActCooLR – High-Level Learning Rate Schedules using Activation Pattern Temperature

## Abstract

We consider the aspect of learning rate (LR-)scheduling in neural networks, which often significantly affects achievable training time and generalization performance. Although schedules such as *1-cycle* offer substantial gains over base-line methods, the effect of LR-curves on the training process is not very well understood. In order to gain more insight into the training process, we combine information theoretic ideas and probabilistic optimization, namely simulated annealing. In more detail, we introduce the *activation pattern temperature*, which (i) captures changes in the non-linear behavior of ReLU networks and (ii) is free of hyperparameters and thus is more interpretable. Examining the training process, 1-cycle simply yields a linear decrease in temperature, reminiscent of successful cooling strategies in simulated annealing. In order to test a causal connection, we devise *ActCooLR*, an automatic LR-scheduler that produces declining temperature profiles. In experiments with various CNN architectures and different image classification data sets, we obtain results that perform favorably or exceed the performance of hand-tuned schedules.

## 1 Introduction

Despite the huge success of deep networks, their training dynamics is still ground for many discussions. Above all, the reason for the good performance of such a simple algorithm as Stochastic Gradient Descent (SGD) remains an open question. As stated by Bengio [4], the learning rate (LR) of SGD is "[t]he single most important hyperparameter and one should always make sure that has been tuned". It is considered to steer the amount of noise that regularizes the optimization [6; 22]. Research spans from practical recommendations, such as best practice learning rate schedules of distinct forms [4] to theoretical models that unveil the implicit regularization of SGD that depends on the learning rate [2; 46]. For instance, many training procedures include a *warm up* phase into the learning rate schedules to adapt training to numerical limitations as well as the distinct behavior of the initial training phase compared to the rest of training [14; 15; 36]. Recent studies divide the whole training process into phases of distinct characteristics. Nevertheless, the number of regimes or phases is still under discussion, most commonly described as two or three phases ([13; 29; 31; 32; 39]). A broad variety of work introduce sharpness based measures that give mathematical characterizations of the loss landscape promising a deeper understanding of the phases, trainability and generalization of deep networks [24]. However, these typically either introduce hyperparameters themselves or describe only a subspace of optimization directions.

In this paper, we are trying to understand the effect of varying learning rates on the training process better. As a central tool, we propose a new measure of learning progress, *activation pattern temperature (APT)*. The key idea is to focus on the "hard" part of optimization, which is the fitting of a non-linear function. We therefore measure changes not by step-size in parameter space but counting changes in *activation patterns*, i.e., testing if the decomposition of feature maps into piecewise

linear regions changes. Due to its independence of changes to the linear mappings, the measure is, unlike the original LR and other simple differential measures in parameter space, more stable under reparametrization.

Through the lens of this measure we analyze the training of convolutional neural networks on image classification tasks using several LR schedules (Section 3.2) and find that the commonly used *1-cycle* scheduler [45] has a very simple behavior, namely an approximately linear decrease during training. It also provides some additional insights into the training dynamic, such as connections between temperature and generalization behavior, and a visualization of phase-boundaries for different learning rates.

Using the analysis, we present a method that adapts the learning rate automatically to match a user-specified target temperature profile throughout training. Effective profiles start at high temperature and decrease monotonically until the activation patterns do not change anymore and optimization becomes purely linear. Correspondingly, we name our method *ActCooLR*. As our method matches the performance of previously hand-tuned learning rate schedules in our experiments, it could be considered as a candidate for an effective and efficient, hyper-parameter free automatic LR-scheduler. The computational overhead is moderate , with only one additional forward-pass.

In summary, our main contributions are (i) the introduction of the activation pattern temperature, which reveals a more uniform view of the effect of LR-scheduling on training and (ii), based on this, an automatic learning rate scheduler that provides accuracy for short training times in a fully automatic way.

## 2    Related Work

Driven by the goal to better understand generalization, the training process of deep network training has been analyzed in a large body of work. We would structure the background as follows:

**Training Phases**: One approach of understanding the training process is to describe it in different phases. An early variant of this idea is the work of Bengio et al. [5], who showed that increasing the intrinsic complexity of data during training can help to improve generalization performance. Several studies identify two training phases: the network trains low-frequency features first, yielding low generalization error and continues to learn high-frequency features in a second training phase that is more susceptible to overfitting [42; 26]. Similar observations have been made in studies that also take the effect of the learning rate into account [29; 31; 32]. A more recent study stated that their "experiments suggest that this [(two training phases)] is not the complete picture" [39]. Others show evidence of three instead of two training phases [13; 31]. However, there appears to be consensus in literature that at the beginning of the training, the activations of a network mainly perform a random walk [13; 20]. More practically, increasing the randomness has been shown to even improve performance (see e.g. [40; 54]), having dropout as a more prominent example [47]. The other widely accepted fact relates to the end of training, where momentum becomes increasingly important [29] as gradients directions simplify [16] and the loss landscape flattens [2]. Hoffer et al. [20] have shown that this comes from the loss landscape getting smoother the farther the weights travel from initialization. More recently, mode connectivity has been used as a tool to check whether a modification in the training process leads to distinct optimization trajectories in the loss landscape and its found minima [14; 23].

**Measures**: There are several studies on the correlation between complexity- and norm-based measures [24]. In particular, generalization improvements from flatness of the loss landscape has been discussed both affirmatively and negatively [11; 46]. Nevertheless, sharpness or curvature based methods have been utilized to improve generalization in practice [10; 12]. Numerous work have included additional regularization into the training process. The angle between the momentum vector and the local gradient has been utilized to construct a statistical test to determine convergence [28]. The value and statistics of the loss have also been used for regularization during training, either by relaxing the softmax loss [37] or by adapting the gradients in order to make constant progress on the loss [43]. Lastly, Raghu et al. [41] proposed a method to measure the layer-wise complexity of a network by computing the Singular Value Decomposition of the activations. This method is similar in spirit to ours, but it measures the intrinsic dimensions in a stationary fashion, excluding the training process in the measure itself. In contrast to competing measures, our measure is hyperparameter-free and does not depend on the setting it is evaluated in. It avoids the complexity of rescaling due to

surrounding layers, which plague many continuous measures, by solely focusing on the discrete activations of a ReLU network.

**Hyperparameter Schedules**: Hyperparameter choice has a strong effect on (generalization) performance and convergence speed. Economically, under fixed training budgets, this leads to a trade-off [8; 49]. In cases of small batch sizes, one crucial invariant control parameter has been shown to be the ratio of learning rate and batch size [15; 20]. SGD has been shown to have an implicit bias towards flat regions theoretically that is reinforced by high learning rates [2; 46]. There is strong empirical evidence that large initial learning rates can help with generalization in over-parameterized networks [31; 32]. However, large networks require a "warm up" phase to prevent divergence of deeper layers [14]. While most schedules let the learning rate approach zero with training time, especially the course of the learning rate in the middle of the training process has not yet been analyzed extensively. Our model suggest here an analogy to annealing schemes in discrete stochastic optimization and provides a holistic perspective on the whole training process. LR scheduling is considered to be directly linked to generalization performance [20; 22; 24; 29; 32]. For instance, a cyclic schedule enables to train networks with a good "anytime performance" [34] and the implicit learning rate schedules that are built into adaptive optimizers such as Adam [27] are topic of current research [1; 36]. However, specific research in learning rate schedules is sparse. Although, correctly tuned, 1-cycle achieves the same accuracy using order-of-magnitude fewer training iterations [45], "large models in NLP and vision use schedules which can be easily resumed" [30], such as "clipped" cosine decay [7] or exponential decay [48]. Recently, research in automatic and adaptive hyperparameter tuning has become more prominent. For instance, automatic tuning of decay time for the exponential decay schedule, [28; 30], and meta-networks designed to predict the learning rate based on learned typical training courses ([21], or more specifically, [9; 51]) have been tested. Another approach includes the learning rate into the optimization process by deriving the loss w.r.t. the learning rate as well [3]. Lastly, weight decay has been focus of discussion related generalization gap between SGD and adaptive optimizers [35; 49]. Also, it has been shown to affect the learning rate scheduling directly when used in conjunction with batch norm: every weight decay step increases the effective learning rate by a multiplicative factor for a constant learning rate schedule [33; 52]. The most similar of the named methods to ours from an optimization perspective is probably that of de Roos et al. [10], where successive training steps are used to estimate change of curvature of the loss function to adapt the learning rate automatically; however it requires additional hyperparameters and continuous re-evaluations of the batch-loss.

# 3 Activation Pattern Temperature (APT)

We base our approach on the view that non-linearity is what makes deep networks actually expressive. Throughout this paper, we restrict ourselves to the non-linear aspects of training and study (the popular) ReLU activation function [38], which switches binarily between two linear states. Non-linearity of whole networks is thus encoded in the way the network is switching between those discrete states in an orchestrated way. We call these binary patterns assigned to data "*activation patterns*". Our idea is very simple: We track the change of activation patterns, by comparing corresponding outputs of ReLU layers for the same input data, and use the neg-log-likelihood of these changes to quantify the "step-size" of the training progress.

## 3.1 Formal Definition of APT

Formally, we consider a feed-forward ReLU network $F_\theta^L$ with parameters $\theta$, that contains $L$ activation layers. Further, let $f^l(x) \in \mathbb{R}^{d_l}$ denote the output of such a layer $l \in \{1, ..., L\}$, for an input batch $x \in \mathbb{R}^{B \times d_0}$ of batch-size $B$.

We now define the *activation pattern* for (ReLU) layer $l$ as

$$M_\theta^l(x) := \left(\text{sign}\left(f^l(x)\right)\right)(x) \in \{0, 1\}^{d_l}, \tag{1}$$

which can be seen as a bit-vector of ReLU activated neurons.

**Training:** Training is performed in discrete steps $t = 0, 1, 2, ...$ At each step, an input batch $x$ is considered and the optimizer computes new parameters $\theta_t$. $\theta_0$ is determined by the network's initialization, and

$$\theta_{t+1} = \theta_t + \lambda_t \nabla_{\theta_t} L(F_{\theta_t}^L, x_t) \tag{2}$$

is the update by a single optimization step for batch $x$ under loss $L$ (which, we assume, is informed of the ground-truth outputs $y(x)$).

In this context, the incremental updates to $\theta_t$ from randomly drawn batches $x_t$ make the temporal sequence $(\theta_0, \theta_1, \theta_2, \ldots)$ a Markov chain; the same holds for the sequence of activation patterns $M_{\theta_t}^l(x)$ over time $t$. Additionally, we model the effect of one optimization step on the activation patters as a Markov chain,

$$
(X, Y) \xrightarrow[\theta_{t+1} := \theta_t + \lambda \nabla_{\theta_t} L_{\theta_t}(X, Y)]{\overset{\text{Network Evaluation}}{\underset{\text{Optimizer Step}}{\rightleftarrows}}} \begin{array}{l} \left( M_{\theta_t}^1 \quad, \ldots, M_{\theta_t}^L \right) \\ \left( M_{\theta_{t+1}}^1, \ldots, M_{\theta_{t+1}}^L \right) \end{array} \longrightarrow \left( T_t^1, \ldots, T_t^L \right) \tag{3}
$$

where $X$ denotes the random variables that chooses examples from any distribution, $Y$ describes its true underlying information that we are interested in, and $T_t^l$ the change in activation patterns. Each stochastic process, symbolized by an arrow, optionally adds uncorrelated noise to the mapping.

**Definition:** We now define the *activation pattern temperature (APT)* on layer $l$ as the self-information of the event that an activation pattern has not changed,

$$
T_t^l(x) := -\log_2 \left( \Pr \left( M_{\theta_{t+1}}^l(x) = M_{\theta_t}^l(x) \right) \right), \tag{4}
$$

where $\Pr$ denotes the probability distribution.[1] This estimates the probability of an activation pattern change over the batch $x$. We also define the *average* activation pattern temperature $T_t$ as the average of $T_t^l$ over all (ReLU) layers of the network. This measure is parameter-free and specifies the amount of non-linear change in a network. Its lower bound is $0$, stating that no non-linear change occurred and only the linear parts of the network could have been changed during that step. The temperature approaches infinity if all activations have changed during a single optimization step.

**Computation:** To measure $T_t^l$, we run the forward-pass of a network twice, recording the activation patterns in the first pass and comparing and accumulating changes in the second. In experiments, we observe that evaluation on a single training batch already give good estimates. Thus, we use the single-batch estimate of APT in the rest of the paper, unless stated otherwise. This allows the measure to be calculated at the cost of only one single additional forward pass of the same batch that has been already used for the previous optimization step.

## 3.2 Training Methods & their Training Dynamics

In this section, we utilize the activation pattern temperature (APT) (Equation (4)) to gain more insight into the non-linear training dynamics affected by the choice of learning rate schedules. As a baseline experiment, we compare the training of ResNet-32 (CIFAR-Variant, [19]) trained on CIFAR-10 with *step decay* learning rate schedule.[2] We study the effect of three modifications to the ResNet-Architecture: ResNet with removed shortcut connections, FixUp [53] (no batch normalization) and Pyramid-Net [17] (linear growing number of filters). In Figure 1, we show the learning rate (top row), the corresponding activation temperature of the first layer $f^1$ (middle row), and the Top-1 validation error (bottom row) for the baseline experiment. In the following, we analyze these plots regarding their temperature profiles, training time and generalization performance.

**APT simplified learning rate description:** During the training process, APT reflects changes of the LR (APT is reduced, whenever LR is reduced). APT starts strictly greater than zero, the actual value depends on the architecture as well as the training hyperparameters, and, during the training process, APT is reduced and approaches zero. Most importantly, 1-cycle schedule (blue curves), shows approximately a linear decrease of APT.

**Temperature cools down bottom-to-top:** Defined as a per-layer temperature, we discuss the APT on a per-layer basis in the APPENDIX. We observe that deeper layers have a higher temperature (we call these *hotter*) in comparison to the first layers in a network (we call these *cooler*). Also, we

---

[1] Note the random variable in the formula does not measure the probability for a neuron change, instead, it counts the probability of an activation pattern change.

[2] All used hyperparameters are listed in the APPENDIX.

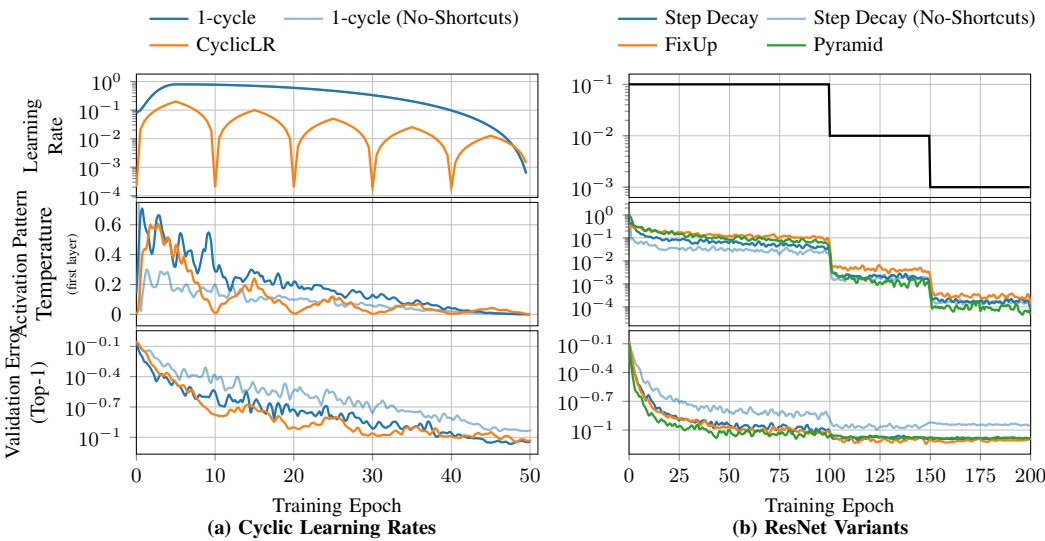

Figure 1: Learning Rates, Validation Accuracy & Temperatures of the first ReLU-Layer in ResNet-32 (and its Variants) on CIFAR-10 for different LR schedules.

observe that in each network the activation pattern temperatures decreases (or "cools down") from bottom to top, in accordance to recent work [41].

**APT possibly relates to generalization:** On the other hand, we observe that the validation error has similarities to the APT profile, despite having no direct relation between the AT and the loss. In this setting, APT is proportional to the validation error. Architectural adaptions also affect performance, negatively (as ResNet in the absence of shortcut connections), and positively (FixUp and PyramidNet) have that characteristics.

**APT is not explained by learning rate alone:** The temperature can change (decrease) despite a constant learning rate. This can be observed, for example, in all experiments of fig. 1(b), especially in the beginning of training, but also with every drop of the learning rate. We conclude: for a fixed setting, the course of APT is not steered by the magnitude of learning rate alone, and APT contains hidden variables responsible for its course.
This is in comply with previous work that studied the early phase of training in more detail [13]. We think that this view could shed some light on the initial phase of training and possibly explains the requirement of using a warm-up schedule in common training schemes beyond numerical instabilities.

## 3.3 Learning Rate Range Analysis

In the following chapter, we want to study the connection between LR and APT. We do so by re-evaluating the exact same update step with a fixed range of learning rates. This allows us to observe the training from a global LR-invariant perspective. In more detail, we carry out this experiment for two different architectures on different data sets, using two different LR schedules. First we train a ResNet-56 on ImageNet, and second a ResNet-50 on CIFAR-10. Both networks are trained using step decay (90 rep. 200 epochs) and 1-cycle scheduler (20 resp. 50 epochs) each. For the initialization and every epoch in training, we freeze the network and re-evaluate the temperature for the exact same range of learning rates. The learning rates we use are uniformly sampled on a log scale, ranging from $10^{-4}$ to $10^2$. The results are shown in Figure 2: The top row shows the average activation pattern temperature $T_t$ for each network and for each LR schedule. The white lines indicate the learning rate, which actually has been used for training.[3] The theoretical temperature is shown in the background as color gradient from black (cold) to yellow (hot). We observe, that after a short initialization phase the temperature has a rather homogeneous behavior

---

[3]For additional plots measuring for the last layer, or another data set, CIFAR-10, also for the hyperparameters used, please refer to the APPENDIX.

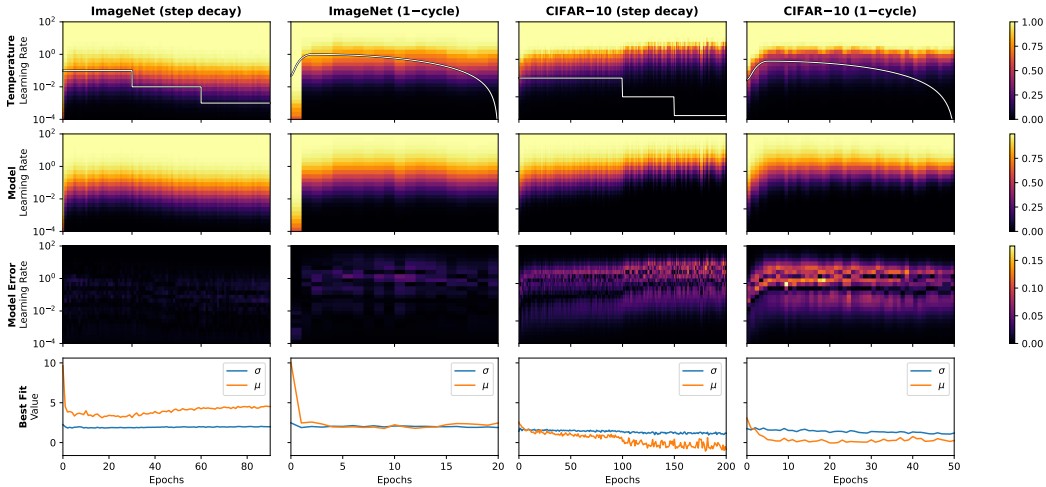

Figure 2: Learning Rate Range Analysis: For each point in training time of ResNet-50 on ImageNet (left two columns) and ResNet-56 on CIFAR-10 (right two columns) each with 1-cycle LR and step decay schedules, we measure the temperature for a theoretical scaling of the global LR ().

over the whole training process. After few initial epochs, we observe only minor changes to the temperature profile. Furthermore, the temperature profiles of ResNet-56 (ImageNet) show a phase in which, from a global and LR-independent perspective, the networks cool down slowly, independently of the choice of learning rate. Most notably, 1-cycle schedule uses LRs in a higher temperature regime (near the vertex) for a longer period of time during training. In contrast, step decay passes through into the colder temperature regime very quickly (ResNet-56/ImageNet) or already stats in it (ResNet-50/CIFAR-10).

## 3.4 Model of the Activation Pattern Temperature

Typically, weight initialization is based on [18], which specifies the initial distribution of the weights in such a way that the output *after* the ReLU-activation is Gaussian distributed on all layers. As the activation itself zeros the output if the weighted sum of normal distributed variables is smaller than zero, we model the probability of a change in activation patterns itself as the cumulative distribution function of a normal distribution. Thus, we postulate the following closed form formula.

**Hypothesis 1.** *The probability of an activation pattern change depending on the used learning rate* $\lambda$ *for a single update step on any layer during training is given by*

$$\Pr\left(M_{\theta_{t+\lambda \cdot \nabla L}}^l = M_{\theta_t}^l\right) = \frac{1}{2} \cdot \left(1 + \operatorname{erf}\left(\frac{\log \lambda - \mu}{\sigma \cdot \sqrt{2}}\right)\right). \tag{5}$$

*The real numbers* $\mu, \sigma \in \mathbb{R}$ *depend on the training process, choice of architecture and layer.*

We show next that the model, given by Equation (5), actually fits in practice. In more detail, for a fixed network state, we fit the model using non-linear least squares to the data shown in Figure 2. We evaluate the fit by visualizing the the evaluated model together with its reconstruction error against ground truth in the second and third row of Figure 2. The estimated parameters $\mu$ and $\sigma$ of Equation (5) are shown in the bottom row of Figure 2. From our experience its values depends on the layer the temperature has been measured in, the data set and architecture used and varies also with the random seed of the initialization of weights.

Most importantly, we could observe favorably only small drifts of the parameter $\sigma$ that defines the width of the Gaussian distribution used in the model. (But we also show an example of larger changes in the APPENDIX). In contrast, the course of $\mu$ seems to be especially affected by the learning rate used for training. We will use this observation and Equation (5) to define an automatic learning rate schedule in Section 4.

# 4 Activation Cooling based Learning Rate Scheduler

## 4.1 Optimization using ActCooLR

The goal of the scheduler is to determine learning rates that impose a specified temperature profile. This has to be done online, adapting the learning rate $\lambda$ to the current network state. As a design decision, we have the option to measure temperatures at every layer, and using adaptive learning rates, even to specify them layer-wise. For simplicity we leave fine-grained adaptation for future work and generally operate with a global learning rate (similar to 1-cycle) and use the mean temperature over all layers for control.

In order to determine the learning rate, we simply rearrange Equation (5) from Section 3.4, which describes the probability of a pattern change using only three parameters: $\sigma$, $\mu$ and $\lambda$:

$$\mu = \sigma \cdot \sqrt{2} \cdot \mathrm{erf}^{-1}\left(2 \cdot P_\lambda - 1\right) - \log \lambda, \tag{6}$$

$$\lambda = \exp\left(\sigma \cdot \sqrt{2} \cdot \mathrm{erf}^{-1}\left(2 \cdot P_\lambda - 1\right) - \mu\right), \tag{7}$$

where $P_\lambda$ denotes the probability given the used learning rate $\lambda$.

Thus, to derive the learning rate $\bar{\lambda}$ required to measure the target temperature $C$, we first estimate $\mu$ using the measured probability $P_\lambda$ using the learning rate $\lambda$ Equation (6). Assuming that the value of $\mu$ and $\sigma$ only changes slowly (see Section 3.3) we can use Equation (7) to estimate the learning rate that would produce the given temperature $C$. We call this learning rate adaption technique ActCooLR. Too reduce computational costs, we limit measuring and readjustments to once every 10 optimization steps. In between two measurements, we just interpolate linearly between the new and old learning rate. The computational overhead is thus moderate, in particular as only a forward pass is needed.

Note, this requires to estimate $\sigma$ in the beginning of training. The value of $\mu$ can be estimated using the formulas given above. We show in the APPENDIX a dynamic version of this technique, that removes the requirement of estimating $\sigma$ and adapting the learning rate dynamically at the cost of additional hyperparameters. A numerical problem arises from temperatures living on a logarithmic scale (Section 3.4). Due to finite sampling, We might estimate a probability of 1 (all patterns have changed) by chance. According to Equation (5) this would correspond to an infinite learning rate. We remove the singularity by an ad-hoc regularizer: For empirical propabilities of 1, we assume that "half an activation has changed", but was not measured.

## 4.2 Designing Target Temperature Curves (CIFAR-10)

Until now we moved the problem of choosing the learning rate curve with a more abstract problem; choosing the temperature curve. It is clear that we want to cool the mean temperature to a value of zero, specifying explicitly that the network shall converge. This temperature is trivially given by a learning rate of 0. We have seen that for sufficiently many data points it becomes increasingly hard to let the network change all patterns with a single optimization step, in the limit this becomes impossible.

Many previous work has analyzed the positive effects of large initial learning rates (see Section 2 for a discussion). A large initial learning rate corresponds directly to a larger temperature. Thus, we assume that we want to start training with a high temperature, i.e. a huge flow of information through the network with every optimization step, and end with a very low temperature (mostly only linear regression to be optimized). Empirically, we observed in the Section 3.2 that 1-cycle shows a linear decrease in temperature. Thus, we test next if a linear decrease in temperature accelerates training by actively controlling the temperature with ActCooLR. As a simple baseline experiment, to test against, we use, again a ResNet-32 on CIFAR-10 and train it using ActCooLR and momentum SGD (see appendix the used Hyperparameters). To test our hypothesis, we define a family of temperature curves of the form

$$C_\gamma^{\mathrm{linear}}(i, i_{\mathrm{total}}, T_{\mathrm{start}}) := T_{\mathrm{start}} \cdot \left(1 - \left(i/i_{\mathrm{total}}\right)^\gamma\right), \tag{8}$$

starting at $T_{\mathrm{start}}$ and cooling down under a reduction factor $\gamma$ to 0.

Figure 3 shows the effect of the selected target temperature curve $C_\gamma^{\mathrm{linear}}$ on the learning rate (Figure 3 (middle row) and the validation error (Figure 3 (bottom row)). From our experience and also in comply with the observations made in Section 3.2, training generally longer on higher temperatures ($\gamma > 1$) achieves favorable performance compared to a faster reduction of temperature ($\gamma < 1$). In

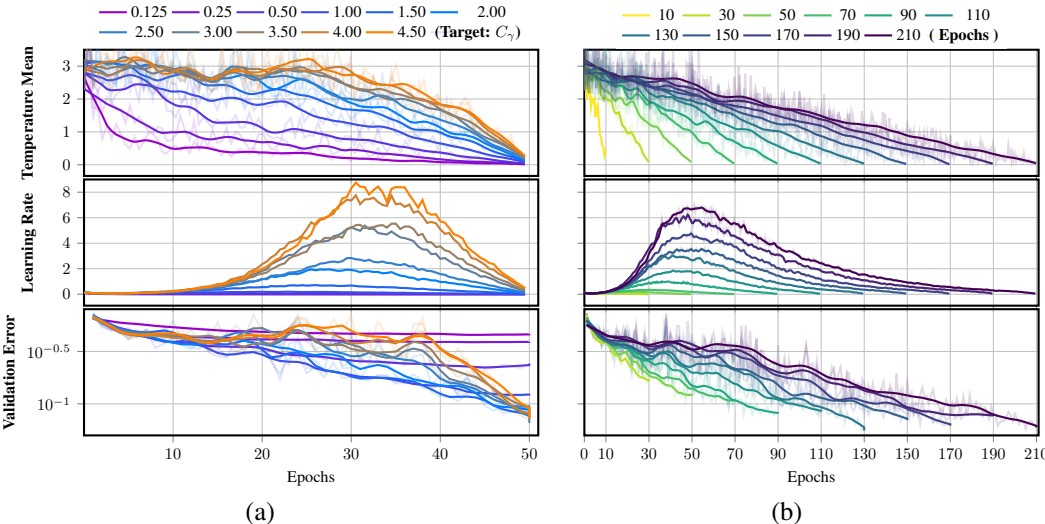

Figure 3: Training ResNet-32 on CIFAR-10 using a linear target temperature curve and (a) using several values for $\gamma$ and (b) for varying number of epochs using a linear temperature curve each. We choose the same initial target temperature for all experiments to be 3.17 (i.e. a 89% chance that an activation has changed), all other hyperparameters, as well as seeds, are kept the same in each experiment set. Top row: measured mean temperature, Middle Row: resulting learning rates, bottom row corresponding top-1 validation error.

case of a too big $\gamma$, the amount of time optimization takes place in a linear way only becomes short at the cost of a worse performance.

We test in the following the same setting for varying number of epochs to show the stability of our method for various training budgets. For simplicity, we restrict the analysis to a linear temperature decay ($\gamma = 1$). In Figure 3(b), we show the disadvantage of our schedule: the validation error remains high for the most number of epochs during training, converging only very late compared to methods like cyclic learning rate or step decay (see Section 3.2). For instance, in contrast to a cyclic learning rate schedules, our method does not show a good anytime performance. On the one hand, cyclic temperatures could also work, but need to be evaluated separately, thus leaving this as topic of future research. We also show in the APPENDIX that our method is independent of the initial learning rate as long as the first few optimization steps do not lead to a diverged network. We believe that finding temperature curves with theoretical bounds is a promising direction for future work to better understand the internal effects. For the rest of our work, we use a linear temperature cool down.

### 4.3 Comparisons with other Methods

Table 1 shows practical results on three different architectures (simple 4-layer CNN, VGG-16 and ResNet-50) on three different data sets (Fashion-MNIST, CIFAR-10, ImageNet). We compare to baselines and two automatic LR-schedulers (ABEL and AutoLRS). The experiments confirm that we reach, similar to our observations for 1-cycle, comparable generalization performance within a restricted training budget (Note though that ABEL uses 200 epochs, unlike the other methods). More results are provided in the APPENDIX.

## 5 Discussion

The key hypothesis of this paper is that tracking the changes to the nonlinear behavior only can provide us with the information needed for step-size control. Our experiments support this view (for image classification with feed-forward ReLU CNNs, which is the class of techniques we restrict our analysis to at this point). Specifically, performing a simple linear decrease in activation pattern temperature already yields a LR-scheduler with performance comparable to 1-cycle, and distorting the temperature curve towards staying longer in the high-temperature regime at the beginning appears to improve generalization performance slightly (at least for the training time scales examined).The

Table 1: Comparison with previous automatic LR-schedulers

| Setup | | | Test error | |
|---|---|---|---|---|
| Data set | Network | Epochs | Method | Top-1 Error |
| Fashion-MNIST | 4-layer ConvNet [50] | 200 | constant LR | 6.95% |
| Fashion-MNIST | 4-layer ConvNet [50] | 200 | ActCooLR | 7.29 % |
| CIFAR-10 | VGG-16 [44] | 350 | step decay | 6.30% [25] |
| CIFAR-10 | VGG-16 [44] | 350 | AutoLRS | 5.87%[25] |
| CIFAR-10 | VGG-16 [44] | 200 | ABEL | 7.1% [30] |
| CIFAR-10 | VGG-16 [44] | 350 | ActCooLR | 6.82 % |
| ImageNet | ResNet-50 [] | 20 | 1-cycle | 27.27% |
| ImageNet | ResNet-50 [] | 20 | ActCooLR | 27.88% |

Table 2: Test errors for Fig. 3.

| Epochs | Test Error (Top-1) |
|---|---|
| 10 | 37.88% |
| 30 | 18.33% |
| 50 | 11.87% |
| 70 | 10.59% |
| 90 | 9.09% |
| 110 | 8.09% |
| 130 | 7.50% |
| 150 | 7.28% |
| 170 | 7.18% |
| 190 | 7.15% |
| 210 | 6.41% |

simple Gaussian two-parameter model of Eq. 5 is approximates the temperature well empirically and leads to a simpler and more efficient automatic LR-scheduling algorithm than a direct optimization of the learning rate. The most important result is probably on the conceptual side: We observe that the rather complex LR-curve of a cyclic (or 1-cycle) scheduler appears to just correspond to an annealing of the APT. This is reminiscent of simulated annealing methods which use a very similar strategy in order to solve combinatorial optimization problems. The logarithmic temperature measure has an analogous form to the temperature in the Boltzmann-distribution of a Markov-Chain-Monte-Carlo (MCMC) optimizer used there. In this sense, our paper reveals that a good LR-scheduler for SGD just performs on the discrete, nonlinear network components a process very similar to simulated annealing. More concretely, we would like to point to the results in Fig. 2 which show a smooth transition between a linear training regime, with low probability of nonlinear changes and a high-temperature, presumably chaotic, regime at the high LRs. By controlling the APT, we can steer training within the band of non-linear, but not chaotic learning automatically, and only dive into the purely linear regime at the end, thereby plausibly obtaining a quicker convergence. The drop in temperature at early training steps is handled automatically, and provides a plausible explanation the utility of initial LW-ramp-up in 1-cycle. It is also interesting that the phase boundary drifts only slowly after this, with constant width, but depends more strongly on the LR-schedule (and data set) used.

**Limitations and future work:** Our consequential findings are empirical; we do not have an analytical derivation of why the training process has the observed properties, or why the proposed temperature curves reach high performance levels. Our empirical observations are consistent over several data sets and rather different CNN architectures. Nonetheless, a broader study on a large corpus of models and architectures, as well as examining applications beyond image recognition and feed-forward CNNs, is an important next step for future work. Further, a predictive theoretical model of the statistical dynamics of activation patterns under parameter trajectories and exploring a closer connection of SGD and MCMC optimization would be interesting avenues for future work.

# Broader Impact

From an application perspective, our paper aims at improvements in learning time scheduling, such that fast-training schedulers similar to 1-cycle can be used with little or no tuning of hyperparameters. To the extend of this being successful, we believe that this has a significant positive impact in saving time and (electrical) energy in the development and deployment of deep networks (aside from potential rebound effects applying to any improvement in efficiency). We advise the reader though to use caution in that our key findings are empirical in nature and there is no proof of absence of negative effects in terms of costs, and/or a random or systematic loss of accuracy (the same applies to related methods, too). The paper should be received as a step forward towards a more efficient automatic training schedules, not as a proven solution ready for wide deployment. We would correspondingly emphasize the impact of the conceptual view of tracking nonlinear optimization and the newly introduced techniques for implementing this over the practical aspects.

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
