# Supplementary Material for:
# ActCooLR – High-Level Learning Rate Schedules using Activation Pattern Temperature

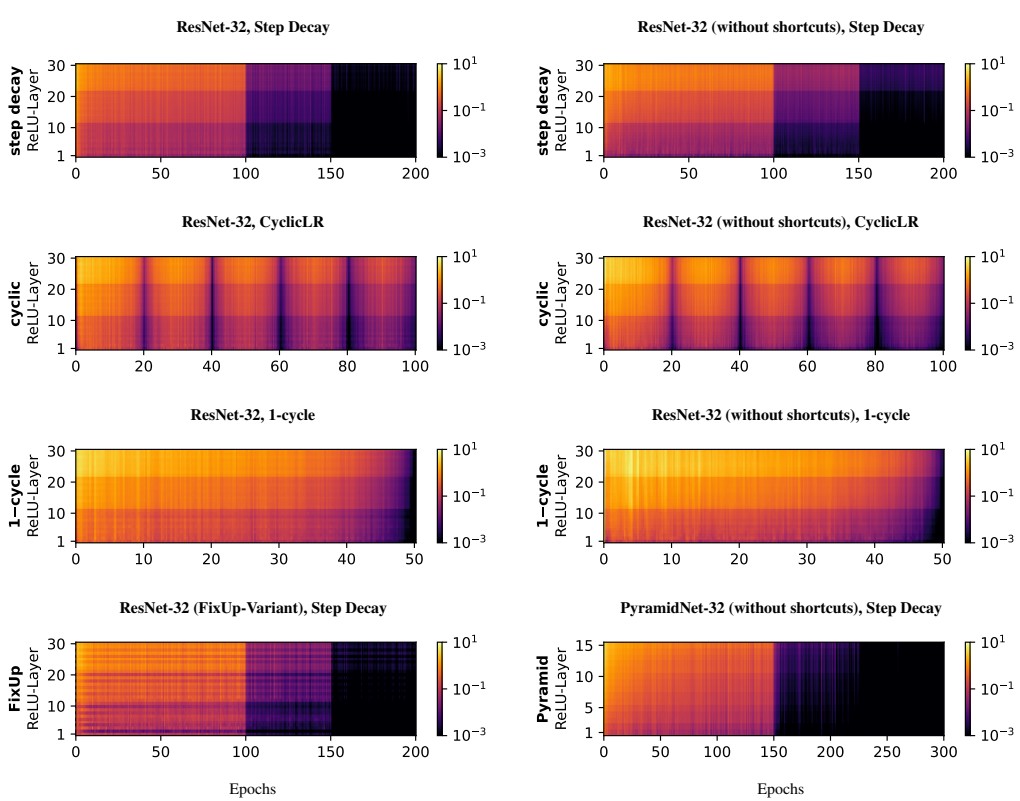

Figure 1: Per-layer temperatures of the whole training process for ResNet-32 on CIFAR-10, using different learning rate schedules (step decay, cyclic and 1-cycle) and architectural modifications to the ResNet-architecture (FixUp, Pyramid, and ResNet in the absence of shortcut connections).

## A   Per-Layer Temperature Curves

The activation pattern temperature (APT) is a layer-wise measure. Thus, in the following section, we analyze the APT of each non-linear block in a network. Figure 1 shows the APT for the full training process (x-Axis) on each layer (y-Axis). In line with Figure 1 of the main paper, we show the temperature profiles of ResNet-32 (and its configurations) trained on CIFAR-10.

Submitted to 35th Conference on Neural Information Processing Systems (NeurIPS 2021). Do not distribute.

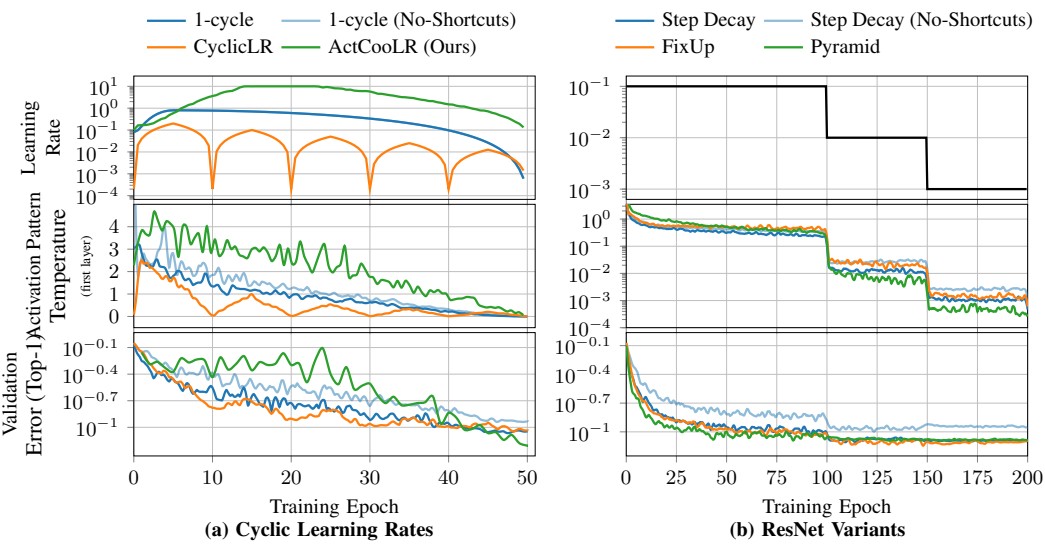

Figure 2: Learning Rates, Validation Accuracy & Mean Temperatures of ResNet-32 on CIFAR-10.

**Temperature cools down bottom-to-top:** In each network of this experimental setup, we observe that the activation pattern temperatures decreases (or "cools down") from bottom to top, in accordance to recent work [4]. Also, architectural choices such as the number of filters affect the temperatures. For instance, the number of filters used are clearly distinguishable by three ResNet-32 blocks: The ResNet-Architecture typically increases the number of filters in each block, which can be seen as increasing base temperatures in each of the plots, as increasing filters also increases the probability of a pattern change. In our experience, we did not found striding to increase the temperature, but only increase the variance of the measurements, probably due to fewer applications of convolutional filters per layer.

**On the Relation of Generalization and AT:** We observe in this experiment that deeper layers generally have a higher temperature in comparison to the first layers in a network. As explained in the main paper, a higher temperature of the first layer seems to relate to generalization, however, we did not found any relation to generalization in the other layers or the mean temperature. See, for instance, the temperature of ResNet-32 with and without shortcut connections in Figure 1; the temperature of the latter are higher despite having a worse generalization performance (compare to Fig. 1 in the main paper).

## B  Robustness of the Choice of Initial Learning Rate

In the main paper we set the initial learning rate to a fixed value of 0.1 (and 0.5 in the case of ImageNet to match the settings used by 1-cycle). The initial learning rate is (most importantly) used for the first few steps to estimate the current temperature of the model. After that, we interpolate to the calculated learning rate according to Section 4.3.

Now, we investigate weather the initial learning rate affects our temperature based optimization. In Figure 5, we test six learning rates that differ each by an order of magnitude (`1.0`, `0.1`, `1e-2`, `1e-3`, `1e-4` and `1e-5`). All other hyperparameters, as well as seeds and the linear target temperature curve starting at `3.03` and decreasing linearly to 0, are kept the same. We observe, that for all learning rates except `1.0` the calculated learning rate profiles as well as validation errors are very similar. For the highest initial learning rate, our method yields a different course of training yielding a better overall network performance in the end of training. As the high initial learning rate increases the temperature in the beginning of training significantly, this result is in comply with our hypothesis that states that training longer on higher temperatures yields better results.

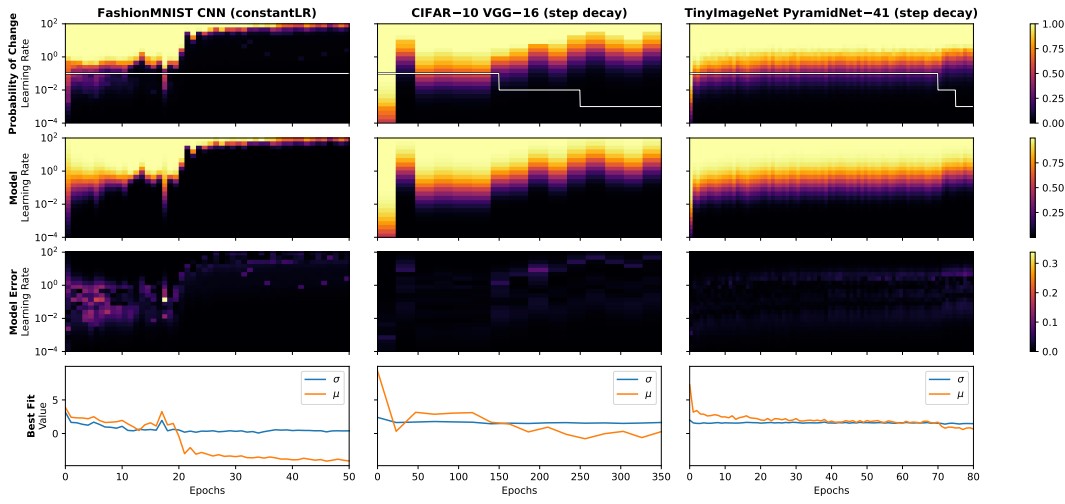

Figure 3: Learning Rate Range Analysis for other data sets and networks.

Table 1: Learning rates used to perform all learning rate range analyses in this work.

| | | | |
|---|---|---|---|
| 0.000630957344480193 | 0.001039122303835169 | 0.001711328304161781 | 0.002818382931264452 |
| 0.004641588833612777 | 0.007644222742526002 | 0.012589254117941668 | 0.02073321573485954 |
| 0.034145488738336005 | 0.05623413251903491 | 0.09261187281287937 | 0.15252229565390182 |
| 0.25118864315095796 | 0.4136820402388507 | 0.6812920690579608 | 1.122018454301963 |
| 1.8478497974222907 | 3.043219887107722 | 5.011872336272725 | 8.25404185268019 |
| 13.593563908785269 | 22.38721138568338 | 36.869450645195734 | 60.72021956909884 |
| 100.0 | | | |

36  It is possible to remove the dependency of the initial learning rate completely by using the estimate
37  of $\sigma$ and $\mu$ to directly calculate the initial learning rate based on a target temperature curve before
38  optimization.

# C  Additional Experiments involving Learning Rate Range Analyses

40  For the learning rate range analysis shown in Section 3.3, respective Figure 2 of the main paper,
41  we additionally measure the activation pattern temperature for a predefined range of learning rates
42  during the training process, while a fixed, predefined learning rate schedule is applied. To reduce
43  approximation errors of the APT itself, we measured the average activation pattern temperature over
44  4096 examples in the case of ImageNet and TinyImageNet, and the full training data set in the case
45  of CIFAR-10 and FashionMNIST. The 25 learning rates (see Table 1) used for the range analysis are
46  in all experiments uniformly sampled in log space between $10^{-4}$ and $10^{2}$. The experiment shown in
47  the main paper shows the measured average pattern temperature over all layers. We show in Figure 3,
48  that our proposed model fits in practice as well as when measured against other setups, i.e we evaluate
49  the fit also with VGG-16 trained on CIFAR-10 and PyramidNet-41 trained on TinyImagenet. In the
50  case of 4-Layer ConvNet trained on FashionMNIST (as described in [7]), we note that the model
51  parameter $\sigma$ changes more significantly (starting at about 3, decreasing to about 0.3). From our
52  experience, another case where the model does not fit in particular well, is whenever a network
53  diverges.

54  We repeat the experiment measuring the activation pattern temperature of the last ReLU-layer.
55  Figure 4 shows that deeper layers have a higher temperature in general, as the last layer starts with
56  smaller learning rates to have high probabilities of all activation patterns changed (of over 90%).
57  Nevertheless, upon inspection of the last ReLU-layer, we observe that training takes place mostly in
58  the non-extreme probability range of $[\epsilon, 1 - \epsilon]$ except for step decay learning rate schedule trained
59  used on CIFAR-10. Note, that when viewed per-layer, the estimate of $\sigma$ remains rather constant.

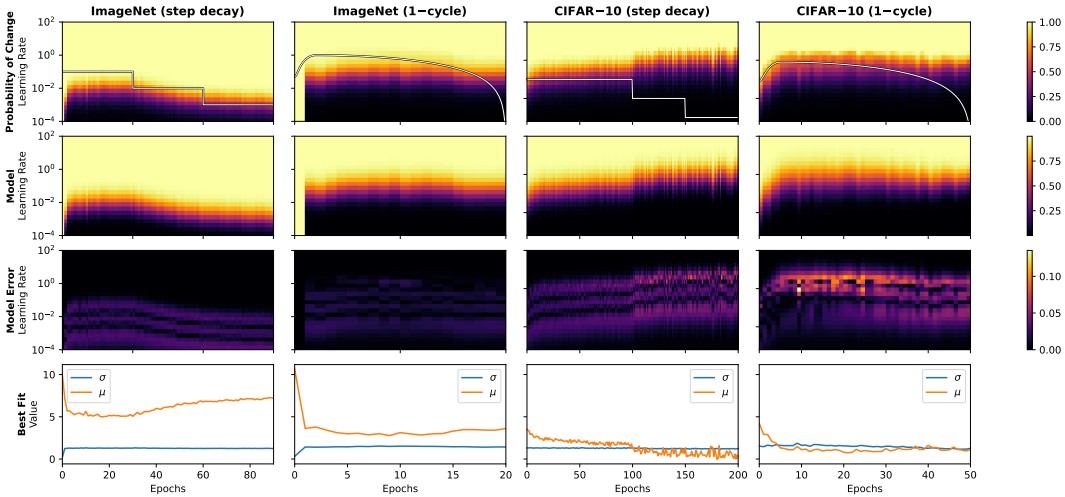

Figure 4: APT analysis (of the *last* ReLU-layer) over a range of learning rates: In contrast to Fig. 2 of the main paper, we show the temperature courses of the *last layer*, given a theoretical learning rate for ResNet-50/ResNet-56 trained on ImageNet/CIFAR-10 using step decay and 1-cycle learning rate schedules.

## D    Optimization using Dynamic ActCooLR

The activation pattern temperature measures the actual non-linear change of a network that represents the convergence of a network's activation patterns directly, i.e. taking the value 0 if and only if no activation patterns have changed. We have seen that the temperature depends (for a fixed network state) monotonically on the learning rate of a single optimization step (see Section 3.4 of the main paper). We approximated the behavior of the temperature w.r.t. the learning rate by a closed form formula (Equation 5) and showed that its parameter $\sigma$ remains approximately constant in many experiments. In the main paper we derived a formula to estimate the learning rate from a predefined temperature, based on the assumption that $\sigma$ remains perfectly constant. Now, at the cost of additional hyperparameters, we discuss an algorithm that removes the requirement to approximate $\sigma$ beforehand. We found that this algorithm works best with the probability of a pattern change, instead of the (logarithmic scaled) temperature.

Assuming we already have a target probability curve $C(i)$, mapping the current training process to a value between 0 and 1 we optimize the learning rates, by specifying first the discrepancy of actual probability to the target probability. We define the temperature error $L_{Temp}$ at time step $i$ as the L2 norm of the difference between the measured probability $\hat{T}$ and the target probability,

$$L_{Temp}(i) := ||\hat{T}(i) - C(i)||_2^2. \tag{1}$$

Similar to the version using our proposed model and the assumption of a static $\sigma$, we either need to specify probability curves and a loss on a per-layer basis, or transform the vector of temperatures for every measurement to a scalar, when we measure a probability for every layer in the network. To simplify the algorithm and the choice of target curve, we optimize the average probability over all layers, instead of all layer-wise temperatures independently.

Knowing that the probability depends monotonically on the learning rate $\lambda$, we can simplify the optimization of Equation (1) by changing the learning rate only slightly into the direction of more/fewer changes. We define $\frac{\partial T}{\partial \lambda} := 1$ and minimize the loss using a separate optimizer, which introduces additional hyperparameters, e.g. a learning rate $\lambda_T$. This compensates for the log-scale given in Equation (5) as well, however, we found that optimizing the learning rate on a linear scale works best. For simplicity, we use SGD for the weight updates, and SGD for the learning rate updates. To reduce computation time, we compute the measured probability once every 10 steps and reapply the calculated gradients on the learning rate in every update step. A fixed learning rate $\lambda_T$ of 0.05 has proven to be a good universal parameter for the probability optimizer. To retain feasibility, we recalculate the probability every 10 steps, reapplying the gradient in the remaining steps without

an estimate of the actual probability of a pattern change. In most experiments we conducted, this method lead to a quick convergence to the target probability at the cost of additional hyperparameters introduced by the SGD loop applied solely to the learning rate of training.

# E   Experimental Setups

We implemented the method outlined in the main paper in PyTorch [3], NumPy [1] and SciPy [6]. Our experiments were run on a DGX-1 (40 Intel Xeon E5-2698v4 CPUs, 512 GB of RAM, 8 NVIDIA Tesla V100-SXM2 16GB GPUs) and DGX-1 (40 Intel Xeon E5-2698v4 CPUs, 512 GB of RAM, 8 NVIDIA Tesla V100-SXM2 32GB GPUs). We used the following image data sets in this work:

- FashionMNIST consists of 60'000 grayscale images with $28 \times 28$ pixels each having ten classes of fashion related articles. Each class consists of 5'000 examples each and 1'000 examples reserved for testing only.[7] For all experiments we apply global color normalization. During training we also apply random horizontal flips. Batch size used is 64. We choose hyperparameters based on a predefined validation set of 1% of the images extracted from the train data set.

- CIFAR-10 consists of 60'000 color images with $32 \times 32$ pixels each having ten classes of natural motives. Each class consists of 5'000 examples each and 1'000 examples reserved for testing only.[2] For all experiments we apply global color normalization. During training we also apply random horizontal flips. Batch size used is 256. We choose hyperparameters based on a predefined validation set of 1% of the images extracted from the train data set.

- ImageNet (ILSVRC 2012, [5]) consists of 1.2 million images belonging to one of $1'000$ classes each. Test accuracies are calculated on the given separate data set consisting of $150'000$ images. We choose hyperparameters based on the performance of a random subset of about 5% ($\approx 64'143$) images extracted from the train data set. For all experiments we perform global color normalization. During training we perform image augmentation, namely random horizontal flips and random resized crops to images of size $224 \times 224$. We perform testing using centered cropping to images of size $224 \times 224$. We use mini-batch sizes of 128 images for training. Additionally, we use label smoothing with weight 0.1 during training. Due to limited resources we restricted the number of epochs to 20, resulting in a single training run time of about 30 hours.

- Tiny ImageNet is a smaller version of ImageNet [5], consisting of $100'000$ images belonging to one of 200 classes. We perform global color normalization to all images, and the same augmentation as for ImageNet, with a crop size of $64 \times 64$. We use batch sizes of 128 for training.

# F   Logarithmic Property of the Activation Temperature

The probability of a pattern change scales logarithmic with the learning rate. It can be shown, that the probability saturates exponentially if we assume that stochastic gradient descent noise affects random pattern changes for large learning rates. Thus, the temperature lives naturally on a logarithmic scale.

**Proposition 1.** *Let $Step(W, b)$ denote the application of computed gradients, scaled with a learning rate $\lambda$. Let further $P_n$ be the measured proportion of signed activation pattern changes on a single layer $f : \mathbb{R}^c \to \mathbb{R}^d$, $f_{W,b}(x) := \mathrm{ReLU}(x \cdot W + b)$ on a given data stream $x_1, \ldots, x_n$, given by*

$$P_n := \frac{1}{n} \sum_{i=1}^{n} \prod_{j=1}^{d} \mathbb{1}_{(\mathrm{sign} \circ f_{W_t, b_t})(x_i)^{(j)}} (\mathrm{sign} \circ f_{Step(W,b)})(x_i)^{(j)}. \tag{2}$$

*In case of background, data or neuron noise, i.e. multiplicative $\mu_{i+j}, \mu_i,$ or $\mu_j \sim \mathrm{Ber}_{1-\epsilon}$ inside the product and increasing*

*(i) number of sampled data points $n$, or,*

*(ii) output dimensions $d$,*

*measuring $\hat{P}_n = 0$ becomes exponentially unlikely.*

Table 2: Test-Accuracies for ResNet-18 and PyramidNet-110 trained on CIFAR-10 using 1-cycle scheduling and ActCooLR. The Hyperparameters are as follows: Momentum $\alpha = 0.9$, Weight decay $5e-5$, Start Temperature $T_0$, Speed of temperature convergence $\gamma$. For 1-cycle we choose the maximal learning rate $\lambda_{\max}$ and the position of the highest point during training in epochs, $t_{\max}$. We fix the momentum scheduling parameters for 1-cycle as follows: $\alpha_{\text{base}} = 0.765, \alpha_{\max} = 0.9$. For ActCooLR we choose a fixed momentum throughout training.

| Data set | Epochs | Network | Method | Hyperparameters | Top-1 Error |
|---|---|---|---|---|---|
| CIFAR-10 | 25 | ResNet-18 | 1-cycle | $\lambda_{\max} = 1.7, t_{\max} = 0.7$ | 7.53% |
| CIFAR-10 | 25 | ResNet-18 | ActCooLR | $T_0 = 4.32, \gamma = 2.0$ | 8.11% |
| CIFAR-10 | 25 | PyramidNet-110 | 1-cycle | $\lambda_{\max} = 3.5, t_{\max} = 0.1$ | 6.3% |
| CIFAR-10 | 25 | PyramidNet-110 | ActCooLR | $T_0 = 2.885, \gamma = 1.0$ | 6.19% |
| CIFAR-10 | 50 | ResNet-18 | 1-cycle | $\lambda_{\max} = 1.9, t_{\max} = 0.15$ | 6.7% |
| CIFAR-10 | 50 | ResNet-18 | ActCooLR | $T_0 = 3.678, \gamma = 5.5$ | 6.94% |
| CIFAR-10 | 50 | PyramidNet-110 | 1-cycle | $\lambda_{\max} = 3.5, t_{\max} = 0.3$ | 5.04% |
| CIFAR-10 | 50 | PyramidNet-110 | ActCooLR | $T_0 = 4.34, \gamma = 10.0$ | 6.17% |

Table 3: Test-Accuracies for ResNet-18 and PyramidNet-110 trained on CIFAR-10 using 1-cycle scheduling and ActCooLR. The Hyperparameters are as follows: Momentum $\alpha$, Weight decay $\eta$, Start Temperature $T_0$, Speed of temperature convergence $\gamma$. For 1-cycle we choose the maximal learning rate $\lambda_{\max}$ and the position of the highest point during training in epochs, $t_{\max}$.

| Data set | Epochs | Network | Method | Hyperparameters |
|---|---|---|---|---|
| FashionMNIST | 200 | 4-layer ConvNet | constant LR | $\lambda = 0.1, \alpha = 0.9, \eta = 0$ |
| FashionMNIST | 200 | 4-layer ConvNet | ActCooLR | $T_0 = 1.25, \gamma = 1.86, \alpha = 0.0, \eta = 0$ |
| CIFAR-10 | 300 | VGG-16 | ActCooLR | $T_0 = 2.423, \gamma = 10.0$ |
| ImageNet | 20 | ResNet-50 | 1-cycle | $\lambda_{\text{init}} = 0.5, \lambda_{\max} = 1.0, t_{\max} = 0.1,$ $\alpha_{\text{base}} = 0.765, \alpha_{\max} = 0.9, \eta = 10^{-5}$ |
| ImageNet | 20 | ResNet-50 | ActCooLR | $T_0 = 4.905, \gamma = 7, \lambda_{\text{init}} = 0.5, \alpha = 0.5, \eta = 0.00001$ |

*Proof.* Suppose we found a learning rate that would change all activations, i.e. yield $T_n = 0$ without the noise term. (Otherwise we would not measure $T_n = 0$ without the noise term). The probability of the noise not changing any sign back is $P(\hat{T}_n) = (1 - \epsilon)^{nd}$ for a small $\epsilon > 0$ that depends on the magnitude of SGD noise. $\qquad\square$

# G   Additional Results & Hyperparameter List

Additionally to Table 1 in the main paper, we list more results using our method ActCooLR with other data sets, or other models in Table 2. For both methods, 1-cycle and ActCooLR, we performed random search of parameters for a fixed number of training time (24 hours each). We chose hyperparameters yielding the smallest validation error and show the test error in Table 2.

The additional experiments support the findings from the main paper – the accuracy obtained from ActCooLR is in a similar range as 1-cycle, on a larger set of different architectures. Please note that neither results reach the best-possible published accuracy values, as non-learning rate related hyperparameters have not been tuned extensively (for both our method and 1-cycle, to obtain a fair comparison).

We provide a list of all hyperparameters used in Table 1 (of the main paper) in Table 3.

# H   PyTorch Implementation

We will release our complete code repository under a free license upon acceptance. In the following, we show the implementation of the key algorithms.

We implemented the method outlined in the main paper in PyTorch [3], NumPy [1] and SciPy [6].
**Measure Statistics**: The *activation pattern temperature* (APT) is measured right within a ReLU-activation. Thus, the following drop-in-replacement of a ReLU-layer enables measuring the APT in

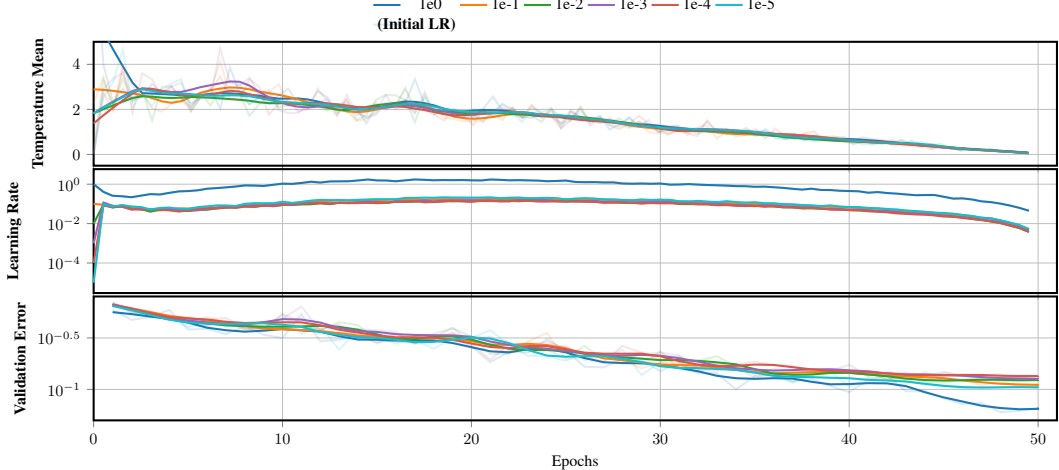

Figure 5: We train ResNet-32 using ActCooLR and vary only the initial learning rate. Parameters used are as follows. Number of Epochs: 50, Start Temperature $T_0 = 3.03$, and Temperature Decreasing Factor $\gamma = 1.0$, SGD Momentum $\alpha = 0.9$.

any model with ReLU-activations. To measure the statistics, the layer requires to be activated using `relu.measure = True`, and the exact same batch needs to be passed twice: once before and once after the optimization step.

```python
import torch
import numpy as np

class APTReLU(torch.nn.ReLU):
    def __init__(self, inplace=False):
        super().__init__(inplace=inplace)
        self.measure, self.counts, self.total = False, 0, 0

    # activate for statistics collection, deactivate else
    def set_active(self, isactive):
        if isactive:
            self.counts, self.total = 0, 0
        self.measure = isactive

    # measure statistics while forward pass
    # note: this asserts channel dimension to be as position 1
    def forward(self, in_tensor):
        out = super().forward(in_tensor)
        if self.measure:  # the same batch needs to be passed twice
            input_sign = out.sign()
            if hasattr(self, "act_before_opt"):  # is after optimization step
                pattern_same = (self.act_before_opt == input_sign).all(1)
                self.counts += pattern_same.sum()
                self.total += (out.shape[0] * np.prod(out.shape[2:]))
                del self.act_before_opt
            else:  # is before optimization step
                self.act_before_opt = input_sign.detach()
        return out

    # ... continued on next page
    @property
```

```
33    def temperature(self):
34        assert self.total > 0
35        if self.counts == 0:
36            self.counts = 0.5 # caps such measurements as log(0) is undefined
37        return - torch.log2(1.0 * self.counts / self.total)
```

The estimation error can be reduced by simply increasing the batch size (which probably affects networks dynamics e.g. batch norm), or alternatively, use the complete (training) data set (with constant mini-batch size). The latter enables a better estimate as it does not distort training dynamics, especially when using layers such as batch norm that includes the statistics of the whole batch, but adds additional memory complexity, because the activation patterns of every mini-batch needs to be saved. Alternatively, copies of both network models, before and after the optimization step, can be saved. This way, the activation patterns of every batch can be computed just-in-time.

**Helper methods**: To convert values between *probability* and *temperature*, we use the following methods.

```
1    def prob2temp(t):
2        return -t.log2()
3
4
5    def temp2prob(t):
6        return 1 - (-t).exp()
```

**Fit model in the initialization point**: Before we can start using ActCooLR in training, we need to fit the model, given in the main paper, to get an estimate for $\sigma$.

```
1    import numpy as np
2    from scipy.special import erf
3    from scipy.optimize import curve_fit
4
5
6    def model_fit(lrs, temps):
7        """Estimate sigma with given learning rates and temperatures.
8
9        Args:
10           lrs (numpy array)  : List of learning rates used to estimate
11                                temperatures in the first optimization step
12           temps (numpy array): List of temperatures measured in the first
13                                step of optimization with the given learning rates
14
15       Returns:
16           float: estimated sigma
17       """
18       assert len(lrs) == len(temps)
19       def fit_p(lr, sigma, mu):
20           return ((1 + erf((np.log(lr) - mu)/(sigma * np.sqrt(2)))) / 2
21       return curve_fit(fit_p, lrs, temp2prob(temps))[0][0]
```

**Adapt Learning Rate to Target Temperature**: Once APT has been measured, we can use the formula provided in the main paper to estimate $\mu$ and recalculate the learning rate according to a predefined target temperature.

```python
import torch
import numpy as np
from scipy.special import erfinv

def adapt_lr_with_stats(relus, target_temp, lr_now, sigma, every_step):
    """Adapts the learning rate based on a given target temperature.
    Args:
        relus (list)        : ReLU-layers of type APTReLU
        target_temp (float)  : target temperature
        lr_now (torch.tensor) : current learning rate
        sigma (float)        : precalculated sigma
        every_step (int)     : the number of steps until next measurement

    Returns:
        torch.tensor         : adapted learning rate
    """
    P_now = temp2prob(torch.tensor(
        [prob2temp(r.counts / r.total) for r in relus]
    ).mean())
    P_target = temp2prob(target_temp)

    # calculate new learning rate based on target temperature
    b = torch.log(lr_now) - sigma * np.sqrt(2) * erfinv(2 * P_now - 1)
    lr_target = torch.exp(sigma * np.sqrt(2) * erfinv(2 * P_target - 1) + b)
    if torch.isinf(lr_target):
        raise ValueError("Target Temperature too high")
    lr_now.grad = 1 / every_step * (lr_target.float() - lr_now.data)
    lr_now.data = lr_now.data * (1 - 1 / every_step) \
                    + 1 / every_step * lr_target.float()
    return lr_now

def adapt_lr_without_stats(lr_now, every_step):
    """Reapply gradient to learning rate in the remaining steps
        without a up-to-date temperature estimate."""
    if lr_now.grad:
        lr_now.data += lr_now.grad / every_step
```