# OpenReview forum: "ActCooLR – High-Level Learning Rate Schedules using Activation Pattern Temperature"
_NeurIPS.cc/2021/Conference — NeurIPS 2021 Submitted_

### Official Review · Reviewer_yjMf · 2021-07-15

**Rating:** 6
**Confidence:** 3

**Summary:**

This papers aims to explore the impact of learning rate schedules in neural network training, through   a newly proposed measure denoted the “activation pattern temperature” which examines the probability that ReLU activations will change between the 0 and linear regions. Thus this measure aims to examine the changes in non-linear behavior of the network. Several learning rate schedules are examined through this lens, and a learning rate schedule based on reducing this temperature is proposed and examined.

**Limitations And Societal Impact:**

The authors clearly discuss limitations and broader impact

**Main Review:**

The idea of defining and estimating notion of temperature and linking it to the learning rate is interesting, and the text provides several experiments that explore the development of the temperature throughout the training of several networks. I think there are several aspects of the text that would benefit from further insight in order to understand the potential utility of the temperature measure.

How the APT is computed is not clear from the “computation” paragraph, could the authors provide an algorithm?

Except the first few epochs, the APT seems to track the LR very closely (as seen in Fig. 1) and the APT distribution seems to change very little (Fig. 2). Are the changes in APT distributions in early training understood or can they be explored more, and can these changes give insights into the early training process? Beyond that early training, APT seems very closely related to LR, what additional insights about the training development can I get from APT over LR? How does the APT relate to the gradient norms? Or how does it affect parameter trajectories?

What is the benefit of using an APT scheduler instead of a LR scheduler? It would seem to require the same amount of schedule design, but more computation to eventually determine the LR is needed with APT. How does designing temperature schedule, i.e. a schedule damping of non-linear network changes, make the training process easier, it seems as abstract as an LR schedule?

How do you know when you are in a chaotic vs non-chaotic non-linear regime, and how can a reader see that controlling APT better navigates these regimes?

Table 1 - why are different method comparisons used for ever experiment? It makes it challenging to compare methods across experiments.

The authors state that only tracking the nonlinear behavior is enough to provide the information needed for step-size control. It was not clear to me how the conclusion was drawn. Are there toy experiments that could show this?

------------------------------
After Author response:

I believe the authors have provided detailed answers to all of my questions, and I believe these answers are sufficient and provide more clarification for me. I believe that ensuring this information is clearly discussed in the paper would be an important update for the text. I think the paper would benefit from a deeper mathematical / theoretical exploration of the ideas presented, but the current empirical analysis is still of value. I have update my scores accordingly.

**Time Spent Reviewing:**

4

---

> ### Author Response · Authors · 2021-08-10
> **Author Response 4**
>
> We would like to thank the reviewer for the constructive feedback.
> For the technical questions and recommendations, we would like to provide answers and further explanations below:
>
> - **Q:** **”How the APT is computed is not clear from the “computation” paragraph, could the authors provide an algorithm?“**
>     **A:** To compute the probability of pattern change, we first save the signed activations (or hashes of the signed activations in case memory has to be saved) during the forward pass of each training step. After the optimization step we repeat the forward pass with the exact same batch and count the number of activation pattern changes. I.e. if any neuron changes for an example, we count it as an activation pattern change. (For the hash-variant, it is sufficient to check if the two hashes remain equal). For Convolutional Neural Networks, we interpret every image patch as one example, resulting effectively in a larger data set. Using the probability of pattern change, the APT can be directly calculated using Eq. 4. As the temperature typically changes only very slowly, it is sufficient to measure the temperature every $n$-th step during optimization. In section H of the supplementary material, we also provide PyTorch code that can be used as a drop-in replacement for `torch.nn.ReLU` layers.
> - **Q:** **“Except the first few epochs, the APT seems to track the LR very closely (as seen in Fig. 1) and the APT distribution seems to change very little (Fig. 2). Are the changes in APT distributions in early training understood or can they be explored more, and can these changes give insights into the early training process?“**
>     **A:** The goal of this work was to describe the entire training process with the help of a unified model, including the early or initial phase of training, the middle phase, which, however, has been little studied yet, and the later phase of training. We validate this claim by fitting our model to the whole training process, including the initialization point which is part of the early phase of training. To answer the question: by inspecting Figure 2, the initialization point of the training also fits our model as well. We admit, however, that a “zoomed-in” version of Figure 2, for instance, would be very interesting as an additional plot in the appendix. In short, while demonstrating that a global model can fit the whole training time is the main result, the ability to automatically and seamlessly handle the initialization phase, where the parameter sensitivity of the network changes most dramatically, is indeed the strongest piece of evidence supporting the utility of the proposed model.
> - **Q:** **”Beyond that early training, APT seems very closely related to LR, what additional insights about the training development can I get from APT over LR? How does the APT relate to the gradient norms?“**
>     **A:** Adding to the answer above: From our perspective, the biggest insights about the training development are that the training dynamics changes only very slowly in nature. The model given in Hypothesis 1 predicts the percentage of an activation pattern changes in the layer or network (depending if the model was fitted to the mean temperature or the layer-wise temperature) given any learning rate *before* the actual optimization step. This is a novel insight to the best of our knowledge and suggests which ranges of learning rates actually perform big or small steps in function space. We believe, this could become important when constructing networks without normalization, new architectures or initialization schemes.
> - **Q:** **”Or how does it affect parameter trajectories?“**
>     **A:** The APT itself is only a passive measure and thus, does not affect the training dynamics when only measured. The potential change in parameter trajectories induced by ActCooLR, however, is a very interesting research question we however did not yet consider.
> - **Q:** **”What is the benefit of using an APT scheduler instead of a LR scheduler? It would seem to require the same amount of schedule design, but more computation to eventually determine the LR is needed with APT. How does designing temperature schedule, i.e. a schedule damping of non-linear network changes, make the training process easier, it seems as abstract as an LR schedule?”**
>     **A:** The big difference between scheduling temperature and learning rates is simplicity: A simple linear decrease in temperature from high (close to maximal) to zero leads to good results. Controlling the learning curves is much more involved: The parameter space has a much more complex metric. For example, as clearly shown in Figure 2, networks are much more parameter sensitive at initialization than during later training. Figure 2 also shows a phase transition from high to low change rates in the non-linear activations. Successful traditional training schemes appear to just cross this transition boundary from high activation change rates to low change rates (linear optimization) in the end. In terms of traditional learning rates, careful parameter tweaking and experimentation is required to identify a stable regime, and non-linear, heuristic curves have to be prescribed in order to navigate the transition area under the changing parameter sensitivity of the network in the initial phases. In contrast, our model yields already good results with a very simple criterion (start at high, but not maximal temperature and linearly anneal to zero). Slight bending of the temperature curve additionally improves the results.
>     In short, the advantage is due to Occam's razor: A model that yields a much simpler explanation for the same result/effect.
> - **Q:** **”How do you know when you are in a chaotic vs non-chaotic non-linear regime, and how can a reader see that controlling APT better navigates these regimes?“**
>     **A:** We name the regime where all activation patterns change at each step (yellow area in Fig. 2) as chaotic, and those where no changes take place (dark area in Fig. 2) as linear. The latter is obvious, as no non-linear behavior changes. The first reflects the fact, that training becomes unstable when veering too deeply into this regime. By measuring temperature (rather than step sizes in parameter space), we obtain direct control over the activation change rates and can avoid instability automatically, while (also automatically) maintaining high change rates (temperatures) during the early training phases, which improves generalization (as we show in our results, and as one would expect from prior work). Without the concept of APT, finding the right range of values requires trial & error.
> - **Q:** **”Table 1 - why are different method comparisons used for ever experiment? It makes it challenging to compare methods across experiments.“**
>     **A:** We tried to match the experimental setups of other works that also adapt the global learning rate dynamically during training. We admit, for the sake of comparability with ABEL, we should also have specified a training run with 200 epochs. We will adapt the table in our next revision accordingly.
> - **Q:** **”The authors state that only tracking the nonlinear behavior is enough to provide the information needed for step-size control. It was not clear to me how the conclusion was drawn. Are there toy experiments that could show this?“**
>     **A:** ActCooLR uses only temperatures, i.e. change rates in the discrete activation patterns, to perform the whole step-size control while yielding results comparable to state-of-the-art learning rate control schemes. This shows the claim conclusively (within the scope of empirically studied examples, of course).

---

> > ### Comment · Reviewer_yjMf · 2021-08-27
> > **Update based on response**
> >
> > Thank you for your detailed and considered responses. Indeed this does clarify some of my questions. I believe a more thorough theoretical exploration of APT would be of value, which may help clarify why we might see some of this behavior. Nevertheless, I believe that this empirical analysis is also of value. I suggest the authors update the text with the clarification provided in the responses and the plots mentioned that would go into the appendix. I have updated my scores accordingly.

---

### Official Review · Reviewer_weBR · 2021-07-16

**Rating:** 3
**Confidence:** 3

**Summary:**

The paper introduces a measure (Activation Pattern Temperature, APT) of the rate of change in the function modelled by a neural network (with ReLU non-linearities) as training progresses.

This measure is then used to set a learning-rate schedule by regulating the APT, an adaptive learning rate schedule denominated ActCooLR.

The paper compares the performance achieved by ActCooLR to a standard step-wise decaying learning rate and to a schedule called 1-cycle.  ActCooLR did not obtain better results than the alternatives.

**Limitations And Societal Impact:**

The authors haver addressed the limitations and potential negative societal impact adequately.

**Main Review:**

The main idea presented in the paper, measuring the rate of change to a different linear regime (per linear-ReLU layer) is original and interesting. However,  the paper does not yet provide enough theoretical insight or compelling practical methods to warrant  being presented to a wide audience. The method, as presented, is only applicable to networks with piece-wise-linear non-linearities, and the adaptive learning-rate introduced does not achieve better results than the alternatives.

The paper starts with a good introduction to related work and the problem of hyperparameter scheduling. But the quality of the exposition deteriorates later on in the paper.  For example:
 * The reasons why the APT measure is denominated a temperature is not clearly explained given that it is introduced as a log-probability in equation 4.
* There are several hints in the paper that the authors will connect this measure to temperatures in simulated annealing but this connection is never explicitly presented.
* Why is APT calculated per layer, and not for the whole network or per unit? This is not even discussed in the main paper.
* The sequences of $\theta$ and $M$ are mentioned to be Markov chains, but this is not necessary for the rest of the exposition. Same when in line 193 the authors mention there are hidden variables responsible for the course of APT, this is rather mysterious and not necessary for the exposition.
* The figure captions are incomplete and it is necessary to read the main text to understand what is being presented. Sometimes the axis labels used are difficult to understand (e.g. Figure 1 for the validation errors, why use $10^{-0.1} - 10^{-1}$ instead of 0.8 - 0.1?
* They introduce Hypothesis 1, and then they support it by saying it fits in practice. This gives a veneer of formality that is not really necessary. Simply saying they are fitting the APT vs learning rate function with a logistic would be clearer.

Finally, ActCooLR does not obtain better final performance than the alternatives (which are simpler to implement and more general).

I would like to end encouraging the authors to pursue the ideas further, as they are interesting but the paper is not yet conference-ready. Some of the ideas left for future work in the paper, like using a different learning-rates per layer, may push the final accuracy and make it a practical alternative for applications.

**Time Spent Reviewing:**

3

---

> ### Author Response · Authors · 2021-08-10
> **Author Response 3**
>
> We would like to thank the reviewer for the constructive positive feedback.
> We address the interesting questions brought up in the review in the following, looking forward to further discussions:
>
> - **Q:** **”The reasons why the APT measure is denominated a temperature is not clearly explained given that it is introduced as a log-probability in equation 4.“**
>     **A:** Originally, we found the expression as an empirical fit. A temperature of $0$ naturally corresponds to no pattern change. The temperature can grow indefinitely, but requires in general indefinite learning rates. Given that a data set can indeed have indefinitely many data points, the cost for every activation pattern in the data set to change could explode in the general network case.
>     From an information theoretical perspective, one could motivate the measure as the amount of information coded in the discrete activation patterns that changed ("was learned") at that optimization step.
>     To explain the notion of "temperature" for the log-probability: Simulated annealing is motivated by physics, where the probability of a state in equilibrium is given by the Boltzmann-distribution $p(E) \propto \exp \left( -\frac{E}{k_\mathrm B T} \right)$. Taking the neg-log of the probability returns a value proportional to the temperature $T$ of the system.
> - **Q:** **”There are several hints in the paper that the authors will connect this measure to temperatures in simulated annealing but this connection is never explicitly presented.“**
>     **A:** We admit, the connection between simulated annealing based on the Boltzmann-distribution to our core idea is not obvious and rather implicit. In simulated annealing, the temperature of the system corresponds to a probability of an explicit state (i.e. positions of particles) of the system. In our case, the corresponding particle states are rather abstract: Implicitly, we define an activation pattern change for one single image patch (in case of convolutional networks) or one single example (in case of fully-connected networks) as a particle movement in the system. It is true, that we did not prove theoretically that Eq. 4 or 5 is correct, which would prove the connection to simulated annealing directly. Instead, this paper *postulates* these equations (assuming that the training dynamics can be interpreted as a physically motivated system in which particles behave in a binary way, where change occurs or no change occurs) and then provides an empirical validation that this model is a good fit to observations. The fact that (in our experiments) the probability of an activation pattern change during the training of neural networks seem to be accurately modeled by the simple, fixed model given by Eq. 5, is, in our opinion, the most surprising result of our work.
> - **Q:** **”Why is APT calculated per layer, and not for the whole network or per unit? This is not even discussed in the main paper.“**
>     **A:** Thanks for the question – this is an important aspect, and a discussion should be added to the next revision. We have indeed considered other variations of APT. For instance, defining APT on the whole network would measure the probability that *any* neuron in the whole network would have changed for one single optimization. In our experiments, we were not able to utilize such a measure effectively, as the condition is met nearly for every optimization step, especially for very deep networks. Additionally, to estimate its value, one would require greater batches to estimate its value accurately. It is known from literature that different layers have different training speeds, thus we restrict ourselves here to the layer-wise measure.
>     We have also tested the other proposed variation of the APT, namely a neuron-wise estimate (using the proportion of same activation per neuron). If two patterns are the same, it yields the same result as the APT. If two patterns differ then the APT counts this as one difference in contrast to the per-neuron-temperature sums over each changed neuron. In our experience, the neuron-wise probability had much lower values throughout training, while the per-layer probability (used by APT) ranges from 0 to (nearly) 1, where training becomes more unstable, and is thus normalized.
>     From our experience, we did not find this measure to be as predictive as the layer-wise APT, also optimization using neuron-wise APT did not achieved the same performance. In general, a better theoretical concept to capture the different scopes of non-linearity (from local to global emergent patterns) is probably an important aspect for future work.
> - **Q:** **”The sequences of $\theta$ and $M$ are mentioned to be Markov chains, but this is not necessary for the rest of the exposition. Same when in line 193 the authors mention there are hidden variables responsible for the course of APT, this is rather mysterious and not necessary for the exposition.“**
>     **A:** Good point, the discussion can be simplified at this point. The reason for explaining a unknown hidden random variable was that a constant learning rate can have a changing APT. Thus, there cannot be a 1:1 formula connecting those two quantities. As can be seen later in the paper, we need additionally $\mu$ and $\sigma$ to explain the model given by Eq. 5 completely.
> - **Q:** **”The figure captions are incomplete and it is necessary to read the main text to understand what is being presented. Sometimes the axis labels used are difficult to understand (e.g. Figure 1 for the validation errors, why use instead of 0.8 - 0.1?“**
>     **A:** Thanks the feedback – we will revise figure captions and axis labels.
> - **Q:** **”They introduce Hypothesis 1, and then they support it by saying it fits in practice. This gives a veneer of formality that is not really necessary. Simply saying they are fitting the APT vs learning rate function with a logistic would be clearer.“**
>     **A:** The wording can of course be adjusted in style. It is pointed out prominently as an empirically verified hypothesis because, as explained above, we believe that observing the fit of Equation 5 is actually the most important contribution of the paper. This formula predicts *before* actual further optimization what number of activation patterns in the whole training data set will change when using learning rate $\lambda$ for a single training step using just a batch of examples. This finding is surprising as it abstracts the complex nature of random activation flips into one fixed model. We believe that this simplification can help building better models of training dynamics (currently, a lot of modeling is restricted to initialization of networks, because a simple, analytical model of the time-dependent process is lacking).

---

### Official Review · Reviewer_bPGR · 2021-07-16

**Rating:** 7
**Confidence:** 3

**Summary:**

The paper attempts to understand the change of activation patterns, meaning the post-ReLu sign flip before and after one parameter update, for the same batch of input data. A measure, activation pattern temperature (APT), is then defined on a layer as the self-information of the event that an activation pattern has not changed. The author uses APT to interpret different training methods and training dynamics. Finally, based on the observation that existing learning rate schedule adheres to a linear decrease of temperature, the author propose ActCooLR, a LR scheduler that was tested on a variety of training tasks.


**Limitations And Societal Impact:**

I strongly suggest rewriting the title (and maybe the abstract and part of the introduction) to reflect that this paper is more about the notion of APT, and how it can be an investigatory tool to uncover unknown bits of model training, than it is about a brand new, game-changing LR scheduler (too bad it's not). And I encourage the author to further explore APT as a tool to study other aspects of learning, not just LRs.

The major limitation, as far as I can see, is that ActCooLR doesn't really perform better than other hand designed schedulers. I do not overly discredit the overall scientific value of this work because of that, but it remains a major drawback of this work.

**Main Review:**

The paper is elegantly written, very engaging throughout. One impression I get after the first read is that it is more about the temperature concept, aka, APT, than the learning rate scheduler ActCooLR. But the title seems to suggest otherwise. One suggestion would be to modify the title to truthfully reflect the content.

Possibly one remark that would be raised by reviewers, and all readers in general, is that the eventually devised LR scheduler doesn't really work -- on all the experiments it didn't show a superiority over other contemporary methods. But I am a strong believer that the merit of a scientific work does not solely depend on the positivity of results. And I do believe the APT method, the relationship between LR and APT, and the empirical analysis provide great insight to the community that's actively trying to uncover the mystery of training dynamics.

**Time Spent Reviewing:**

3

---

> ### Author Response · Authors · 2021-08-10
> **Author Response 2**
>
> We would like to thank the reviewer for the detailed and positive feedback.
> We agree that it would be a good avenue for future work to extend our research in the proposed direction, bringing discrete optimization and neural network training even closer together.
>
> It is true that the presented ActCooLR scheduler has no edge in performance over existing LR schedulers. The main appeal is its conceptual simplicity, where the whole LR-control behavior emerges from a single principle (simple temperature decrease) rather than from multi-phase, hand-tuned periods. This might also offer perspectives for more easy to use practical schemes, but the focus of our paper is at this point on the conceptual side.
> Admittedly, that the title of the paper might be confusing in (unintendedly) suggesting too much effort on a newly developed method (which is only a vehicle towards understanding training dynamics at this point, not a plug-in replacement for existing schemes).
>
> Our goal in this work was indeed, as pointed out in the review, to abstract the concept of learning rates by providing evidence that the step size used for optimization of neural networks actually may perform mainly a discrete search in function space. The presented method, ActCooLR, uses this idea to show that this abstract concept is already sufficient to train neural networks to a similar quality using less hyperparameters.
> Instead of a complex curve, we replace the choice of learning rates at any point in training time with a target start temperature and a decreasing factor. Interestingly, by adapting the learning rate automatically, the presented method achieved training at high learning rates that are often considered to be beneficial for generalization (see related work).

---

### Official Review · Reviewer_gCvn · 2021-07-16

**Rating:** 6
**Confidence:** 3

**Summary:**

This work contributes a analysis of the dynamics of neural network training using a new quantity called the activation pattern temperature, that measures the amount of (ReLU) neuron activations that switch from 0 to positive, or positive to 0.

They measure the APT during training of typical CNNs using different learning rate schedules and show that using these schedulers neural networks typically undergo a transition from a high temperature regime to a small temperature one.

They then design a learning rate scheduler that adapts the learning rate so that the temperature of the optimization dynamics during training follows a predefined curve -- in this case it decreases linearly from 3.17 (why?) to 0.

**Limitations And Societal Impact:**

yes

**Main Review:**

I enjoyed the originality and clarity of this paper. There are however a few points that can be improved, and I am not exactly convinced that introducing the APT gives a useful tool to analyze training dynamics, or in other word that this contribution is significant enough for a publication at NeurIPS.

(Major) things that can be clarified

1. What new insight can APT give that other quantities in the literature cannot?
2. Additionally, how can we use the APT to design better neural networks?
3. How did you come up with the formula in eq. 5?
4. line 255, you write "we interpolate between the new and old learning rate" => Can you make this statement more precise? At first read it looks that you can anticipate on the future (i.e. the new) optimal values for mu?
5. Are you using momentum in the experiments in figure 2? If so, how does it affect the actual step size in parameter space after applying momentum?
6. In Figure 3., you are using momentum. It seems that in this case, the actual APT is not fully determined by the learning rate, but also by the additional momentum term in the update. Can you please discuss how momentum interacts with the APT?

(Minor) questions relating to log scales in plots:
 1. I question the choice of a log scale in Fig. 1 right for the APT plot. Aren't we looking at the log of a quantity that is already a log (the temperature)?
 2. On the same figure, I find it a little bit weird to use a log scale for the validation error, whereas I have always seen linear scales in other works.
 3. Most of the change in accuracy typically occurs during a few training epochs in the beginning, can you maybe reflect that by plotting epochs on a log scale in Figure 2 e.g.?
 4. In Figure 3, can you use a log scale for the learning rate, as it typically ranges over several orders of magnitude?

**Time Spent Reviewing:**

4.5

---

> ### Author Response · Authors · 2021-08-10
> **Author Response 1**
>
> We would like to thank the reviewer for the very valuable feedback.
> We would like to address the questions brought up in the review. We will also address these issues in the next paper revision, including all editorial remarks/typos.
>
> **Major:**
> - **Q:** **”... in this case it decreases linearly from 3.17 (why?) to 0.“**
>     **A:** We found training with a high initial APT performs better. Too high values, however would result in an near infinite learning rate according to Equation 5. Thus, we have chosen the temperature in such a way that the target APT is achievable during the first few steps of training.
> - **Q:** **”1. What new insight can APT give that other quantities in the literature cannot?“**
>     **A:** The expressivity of neural networks comes from their non-linearity, as sequences of linear layers could be merged into one single linear layer without loss of expressivity. The APT measures the probability of non-linear changes in a neural network explicitly. In contrast to the measures presented in other works, such as gradient- or sharpness-based measures, the APT measures for each layer, how many different activation patterns have changed over the training data set, thereby capturing the non-linear behavior. This can also been seen from a perspective of isolating discrete from continuous behavior, which provides an analogy with simulated annealing methods. One could argue that, of course, one could define plenty such discrete measures on neural networks. The utility of APT comes from its predictive value: It shows that complex learning curves can be interpreted as simple linear decrease in the rate of discrete changes to the non-linearities. Further, it explains the complexities of the initial training phase as a simple transition from high to lower parameter sensitivity; in the APT-metric, the initial training phase can be treated uniformly with the overall training time.
>
>     As a consequence, APT is, to the best of our knowledge, the first model that gives a qualitative explanation for the rather complex learning rate curve of the 1-cycle scheduler, namely a linear decrease of APT. Conversely, enforcing a linear decrease of temperature (using ActoCooLR) results in similar learning rate curves (see Figure 3) and may even self-regularize training towards very high learning rates (of about 8 in the case of Figure 3). While high learning rates are often considered to be beneficial for generalization (see related work) the learning rate that is chosen by ActCooLR may be greater by an order of magnitude compared to the learning rates used for similar experiments from other studies (without stability risks due to the adaptive nature of the approach).
> - **Q:** **”2. Additionally, how can we use the APT to design better neural networks?“**
>     **A:** While APT, in practical terms, is a computationally cheap practical tool for monitor training of neural networks and might help in automating learning rate control more effectively, our paper does not address this network design and architecture at this point. Further, to our understanding, it also does not directly imply concrete suggestions in that regard.
>     Nonetheless, APT might be a valuable tool for a better understanding of information propagation that might be useful for future work addressing architectural improvements: The activation-layer wise measurement of APT indicates which layers in a neural network experience more non-linear change (greater APT measurement) or fewer non-linear change (smaller APT measurement) than other layers in the network. As can be seen in Figure A.1, later layers undergo greater non-linear change that the first layers in a network. Conversely, this means that towards the end of the training, large parts of the network no longer undergo many non-linear changes. This however, could be important when trying to exploit the whole range of expressivity of a neural network. Our intuition is that the APT can help to find better neural architectures with a better information theoretic throughput during training. For instance, it may help to construct networks without normalization. Such endeavors often struggle with a lack of independence from the scale at which the gradient updates are applied. The APT, on the contrary, directly measures the effective step size of the non-linear function space component inside the neural network.
> - **Q:** **”3. How did you come up with the formula in Eq. 5?“**
>     **A:** From Figure 2, we found that the probability of a pattern change follows an S-shaped curve, where no change naturally occurs for the learning rate of 0. The probability of no pattern change (i.e. all patterns remain equal) models whether the decision boundary of (initially) Gaussian distributed weight changes for the input data after one single step of optimization changes. Especially research that analyzes the initialization of neural networks assumes also the individual data dimensions to be approximately Gaussian distributed. Our intuition was that the change of sign of the pre-activations in the first optimization step is then also Gaussian distributed, where the probability of a change for a specific data point becomes more likely the more the weights are changed. In this paper we validate this intuition by showing that this model fits in practice over the whole training run for typical training runs on CIFAR-10, ImageNet, TinyImageNet and FashionMNIST.
> - **Q:** **”4. line 255, you write "we interpolate between the new and old learning rate" => Can you make this statement more precise? At first read it looks that you can anticipate on the future (i.e. the new) optimal values for mu?“**
>     **A:** The idea is to reduce the single additional forward pass to compute the APT by measuring it only every 10th batch. We adapt then the learning rate in this step and in the remaining 9 batches by applying one tenth of the way towards the calculated optimal learning rate according to Eq. 5. Thus, the learning rate we calculate lags behind, but considering that $\mu$ changes only slowly during training (see Figure 4), we did not found this to be a drawback.
> - **Q:** **”5. Are you using momentum in the experiments in figure 2? If so, how does it affect the actual step size in parameter space after applying momentum?“**
>     **A:** We include momentum in all experiments (see also the hyperparameter lists for the respective experiments in the supplementary material). In more detail, we use the PyTorch variant of SGD with momentum, thus applying the learning rate directly to the adapted momentum buffer of each optimization step. Our method adapts that learning rate directly, leaving the momentum factor constant.
> - **Q:** **”6. In Figure 3., you are using momentum. It seems that in this case, the actual APT is not fully determined by the learning rate, but also by the additional momentum term in the update. Can you please discuss how momentum interacts with the APT?“**
>     **A:** We too expected that especially a higher momentum factor could affect the optimization of APT. However, the connection is not obvious: Even for higher momentum factors, the learning rate still steers the step size used for the optimization step. (I.e. a learning rate of $0$ always corresponds to a temperature of $0$). Our measure does not take the direction of gradients into account. Thus, going into the negative direction of the gradient should, in theory, yield the same probability respectively the same temperature. To conclude, we did not observe a direct connection between higher momentum factors and changes of the APT in our experiments. We think, however, that including also the direction of gradients into such a discrete measure could be an interesting direction of future work.
>
>
> **Minor:**
> - **Q:** **“1. I question the choice of a log scale in Fig. 1 right for the APT plot. Aren't we looking at the log of a quantity that is already a log (the temperature)?“**
>     **A:** This is correct, we could have named the plot ”Activation Pattern Probability“. In order to simplify the amount of notations, we have used the equivalent designation APT on a log scale in this place.
> - **Q:** **”2. On the same figure, I find it a little bit weird to use a log scale for the validation error, whereas I have always seen linear scales in other works.“**
>     **A:** The differences in performance for different schedulers are quite small for a typical runs of ResNet-32 trained on CIFAR-10 and varying schedulers. Thus, we have chosen the log scale solely to improve readability of the figure.
> - **Q:** **”3. Most of the change in accuracy typically occurs during a few training epochs in the beginning, can you maybe reflect that by plotting epochs on a log scale in Figure 2 e.g.?“**
>     **A:** Yes, this could be for example added to the supplementary material to highlight the behavior in the early phases (which indeed has been of identified as an important problem in a lot of earlier work [1]).
> - **Q:** **”4. In Figure 3, can you use a log scale for the learning rate, as it typically ranges over several orders of magnitude?“**
>     **A:** We will change this in our next revision.
>
> [1]: Jonathan Frankle, David J. Schwab, Ari S. Morcos: The Early Phase of Neural Network Training. ICLR 2020

---

### Author Response · Authors · 2021-08-10
**Overall Comment**

First of all, we would like to thank all of the reviewers for their their effort – the very detailed and constructive feedback and suggestions is very helpful in improving our work.

**Next, a small editorial remark**:
While reading our submission, we noticed that Equation 5 should start with $P(... \neq ...) = $, not $P( .... = ... ) = $ (which was probably obvious from the discussion in the text, which describes this correctly). This will of course be fixed in the next revision.

**Then, as final general remark, we believe that it is important to clarify the overall positioning of our work**.
In hindsight, we recognize that the paper could have been clearer in spelling this out (the prominent appearance of the methods name in the title might have been unintendedly contributing to the confusion).

A main issue of concern, pointed out throughout most of the reviews, is that the proposed method *ActCooLR* does not surpass the generalization performance of previous state-of-the-art in learning rate schedulers (i.e., in particular the 1-cycle scheduler) but achieves only similar performance levels. This is correct – the practical advantage of the proposed method is not a performance improvement. The main benefit of the method is its conceptual simplicity, while still being able to match state-of-the-art performance.
The paper demonstrates that previous performance can be obtained by a control scheme
 - that only measures change rates in discrete activation patterns
 - and uses a very simple temperature schedule (rather than complex learning rate curves).
This scheme covers the full training process, from initialization (where we show in experiments that it is much more sensitive to parameter changes) to convergence.

The main contribution of the paper is thus not so much a practical one. With additional future work, ActCooLR could be the basis of an easier-to-use LR-control scheme, but this aspect is not in focus of this paper. The main insight is conceptual, providing a better understanding of the training dynamics, as detailed below:

**Insights on training dynamics:**
Work on the training phases of learning and several measures have already been proposed in the literature before, but, frameworks that unify the choice of learning rate schedules and that put them into context of training phases did not exist previously.
Our approach appears to provide a broader picture by focusing on the non-linear aspects of training, neglecting linear model changes.

**The role of "Equation 5":** In addition to the certainly non-trivial fact that such a simple model is sufficient to reach SOTA LR control, and thus pointing out that these aspects play a crucial role for optimization, we make one more observation that we found very surprising and consider probably the most interesting empirical contribution:
By observing APT during training, we find that the dynamics, viewed in the more invariant APT measure, appears rather stationary: The $\sigma$ parameter in Eq. 5 appears fixed over training, and $\mu$ can be estimated easily using the APT and the used learning rate to measure the APT. (Counterexamples to this stationary nature are from our experience only diverging networks or very small networks, as described in the appendix). This view reduces the choice of learning rate at any point in training to the choice of a probability of pattern changes to a single stochastic model given by Equation 5 ("Hypothesis 1"). The aforementioned results on successful LR-control through the ActCooLR scheme are necessary to appreciate the observation, as this demonstrates that this simple behavior in the APT model is also predictive/decisive for training success.

**In a single sentence:** Our paper shows that step-size effects and control can be explained by a very simple model when focusing on non-linear model changes.

---

### Decision · Program_Chairs · 2021-09-27

**Decision:**

Reject

**Comment:**

In this paper the authors study the rate of sign flips of the RELU units (or more precisely their inputs) during training, and design a learning rate schedule which tries to achieve a linear decrease in this rate over iterations (which was seen empirically for other well-performing LR schedules).

The reviewers found the main idea of studying sign flips for RELUs during optimization to be original and interesting. Their main complaints are that the theoretical explanations for the method are missing or vague, and the experimental results aren't very exciting. Since this method doesn't actually outperform the best LR schedules, its utility comes down to whether it truly offers a simplifying explanation to the problem of tuning LR schedules, and whether this allows it to generalize in non-trivial/surprising ways to new problems. Unfortunately, the reviewers didn't really seem convinced by the explanations, and the experiments were limited to very standard problems that were of the same sort as those used to help design the method in the first place.